# A 1-Hydroxy-2,4-Diformylnaphthalene-Based Fluorescent Probe and Its Detection of Sulfites/Bisulfite

**DOI:** 10.3390/molecules26113064

**Published:** 2021-05-21

**Authors:** Qing Shi, Ling-Yi Shen, Hong Xu, Zhi-Yong Wang, Xian-Jiong Yang, Ya-Li Huang, Carl Redshaw, Qi-Long Zhang

**Affiliations:** 1The Key Laboratory of Environmental Pollution Monitoring and Disease Control, School of Public Health, Ministry of Education, Guizhou Medical University, Guiyang 550004, China; shiqing5900@126.com (Q.S.); xuhong@gmc.edu.cn (H.X.); 2School of Basic Medical Science, Guizhou Medical University, Guiyang 550004, China; shenly@stumail.nwu.edu.cn (L.-Y.S.); zql810921@163.com (Z.-Y.W.); yangxianjiong@126.com (X.-J.Y.); 3Department of Chemistry, University of Hull, Cottingham Road, Hull HU6 7RX, UK; c.redshaw@hull.ac.uk

**Keywords:** fluorescent probe, sulfite/bisulfite, crystal structure, real sample detection

## Abstract

A novel 1-hydroxy-2,4-diformylnaphthalene-based fluorescent probe **L** was synthesized by a Knoevenagel reaction and exhibited excellent sensitivity and selectivity towards sulfite ions (SO_3_^2−^) and bisulfite ions (HSO_3_^−^). The detection limits of the probe **L** were 0.24 μM using UV-Vis spectroscopy and 9.93 nM using fluorescence spectroscopy, respectively. Furthermore, the fluorescent probe **L** could be utilized for detection in real water samples with satisfactory recoveries in the range 99.20%~104.30% in lake water and 100.00%~104.80% in tap water by UV-Vis absorption spectrometry, and in the range 100.50%~108.60% in lake water and 102.70%~103.80% in tap water by fluorescence spectrophotometry.

## 1. Introduction

Levels of anions including organic and inorganic anions (such as sulfurous acid root, amino acids, etc.) are of interest to the field of food analysis, yet they widely exist in food, and also can be used as food additives by being externally added to the food. Moreover, during the food production process, contamination can accidentally arise during one or even several stages, which could result in excessive amounts of anions in the food. These anions not only affect the color, aroma, taste and other qualities of food, but can also play an important role in the health effect of food. In order to evaluate food quality and safety quickly and accurately, it is necessary to carry out qualitative and quantitative analysis of anions with the help of effective analytical testing methods. This is required in order to provide a scientific basis for food production technology, food storage management and monitoring, as well as adherence to the corresponding rules and regulations [1].

Among the differing kinds of anions, sulfite/bisulfite anions (SO_3_^2−^/HSO_3_^−^) plays a crucial role in food preservation due to their characteristics of anti-oxidation, anti-corrosion and enzyme inhibitor, and so are often widely used as food additives in the food industry [2,3]. One of the main atmospheric pollutants is sulfur dioxide (SO_2_) in the physiological environment, and this results in sulfite in treatments with an aqueous base. Large doses of sulfite are toxic to humans and animals and can readily cause adverse reactions and diseases, allergies and severe skin irritation, as well as respiratory problems such as asthma, coughing and gastrointestinal disorders [4,5,6,7]. Other issues include diarrhea, headaches, hypotension, lung cancer and a variety of nervous system diseases [8]. Thus, the amount of sulfite in many countries is strictly controlled and standards are set by the likes of the Food and Agriculture Organization (FAO)/World Health Organization (WHO). JECFA announced that the acceptable daily intake should be less than 0.70 mg/kg [3], and therefore a method for the rapid and sensitive detection of SO_3_^2−^/HSO_3_^−^ in solution would be highly desirable for environmental monitoring, and would also have practical value in the detection of biomedical food safety [9,10]. At present, methods for the detection of SO_3_^2−^/HSO_3_^−^ mainly include ion chromatography/electrochemical methods such as capillary electrophoresis fluorescence [2,11]. According to the Chinese “Standards for the Use of Food Additives” GB/T5009.34-2003 “Determination of Sulfites in Food” colorimetric method, sulfur dioxide in food after extraction should be reacted with detection reagents to generate colored compounds, with a detector at 550 nm for the determination of its absorbance, and a certain range of absorbance is proportional to its content. The detection limit was 4.18 μM. When compared with the traditional method of measuring the sulfite, the fluorescent probe detection method has great potential because of its high sensitivity, high selectivity, non-destructive detection and in situ visualized detection [12,13,14,15,16]. In recent years, the fields of medical biochemistry analysis and environmental monitoring have received widespread attention [17], with water, food and biological systems being subject to study with powerful visual detection tools for anions [18,19,20,21].

In this research, we develop a fluorescent probe **L** which exhibits a good recognition performance and anti-interference ability. It can detect SO_3_^2−^/HSO_3_^−^ in a water environment using UV-Vis absorption spectroscopy and fluorescence spectroscopy.

## 2. Materials and Methods

### 2.1. Equipment and Reagents

The equipment we used included: an Inova-400 MHz NMR Spectrometer (Varian Company, Palo Alto, CA, USA); a VGT-2227QTD type ultrasonic instrument (Shenzhen Gute Hongye Machinery Equipment Co., Ltd., Shenzhen, China); a CP214 Electronic Balance (Shanghai Aohaus Instrument Co., Ltd., Shanghai, China); a Cary Eclipse type fluorescence spectrophotometer (Varian Company, Palo Alto, CA, USA); a UV-visible spectrophotometer of UV-2600 (Suzhou Dao Jin Instrument Co., Ltd., Suzhou, China); a pH meter of pHS-25 (Chengdu Century Ark Technology Co., Ltd., Chengdu, China); and a Bruker Smart Apex single crystal diffractometer (Bruker AXS Company, Karlsruhe, Germany).

1,3,3-Trimethyl-2-methyliminoline, 1-naphthol, hexamethylenetetramine, trifluoroacetic acid, ethyl acetate, methanol, ethanol (EtOH), hexane, dimethyl sulfoxide (DMSO), hydrochloric acid (HCl), anionic metal ions and amino-containing small molecules such as cysteine (Cys) are commercially available and were purchased from Aladdin reagent co., LTD. (Shanghai, China). All chemicals were of analytical grade and were used without further purification. Ultrapure water of 18.2 MΩ cm resistivity was obtained through a water purification system (Youpu Super Pure Technology Co., Ltd. Sichuan, China) and was used in all experiments.

### 2.2. Synthesis of the Compound ***1a***

One gram (6.90 mmol) of 1-naphthol and 1.94 g (13.80 mmol) of hexamethylenetetramine were dissolved in 10 mL trifluoroacetic acid and stirred at 85 °C for 1 h. After cooling, 10 mL of concentrated sulfuric acid diluted to 33% concentration was slowly added into the mixture, and reflux was continued for 1 h. Then, the mixture was twice extracted with ethyl acetate, washed with brine and then dried with anhydrous magnesium sulfate. Filtration, followed by column chromatography separation (*n*-hexane/ethyl acetate = 7:3, *v/v* as eluent), afforded a yellow solid (2.09 g) with a yield of 71%, and the molecular formula of compound **1a** is C_12_H_8_O_3_.

### 2.3. Synthesis of the Fluorescent Probe ***L***

In this process, 0.20 g (1 mmol) of compound **1a** and 0.17 g (1 mmol) of 1,3,3-trimethyl-2-methylene indoline were mixed in 40 mL anhydrous ethanol and stirred at 85 °C for 8 h, and then concentrated under reduced pressure, and separated using column chromatography (*n*-hexane/ethyl acetate = 7:3, *v/v* as eluent) to obtain a bright green powder (0.18 g) in 50% yield. The molecular formula of the fluorescent probe **L** is C_24_H_23_NO_2_. ^1^H NMR (600 MHz, CD_3_OD): δ 10.21 (s, 1H), 10.04 (s, 1H), 9.20–9.22 (d, *J* = 12 Hz, 1H), 9.00–9.07 (d, *J* = 42 Hz, 1H), 8.00–8.02 (d, *J* = 12 Hz, 1H), 6.92–7.92 (m, 4H), 6.68–6.59 (d, *J* = 61H), 5.81–5.83 (d, *J* = 12, 1H), 5.33 (s, 1H), 3.90 (s, 1H), 2.75 (s, 3H), 1.35 (s, 3H), 1.35 (s, 3H). ^13^C NMR (151 MHz, CDCl_3_) δ 191.80, 191.20, 153.40, 137.10, 131.78, 128.89, 128.71, 127.71, 126.53, 126.26, 125.42, 125.14, 125.05, 107.16, 107.11, 102.86, 28.94, 28.72, 28.48, 25.90, 20.13. HRMS calculated: 356.1645, found 356.1650.

### 2.4. X-ray Crystallography

Crystallographic data for ligand **L** were collected on a Bruker APEX 2 CCD diffractometer with graphite-monochromated Mo Kα radiation (λ = 0.71073 Å) in the ω scan mode [21]. The structure was solved by a charge flipping algorithm and refined by full-matrix least-squares methods on F2 [22]. All esds were estimated using the full covariance matrix. Further details are presented in Appendix A. CCDC: 2059923, **L**. These data can be obtained free of charge from The Cambridge Crystallographic Data Centre via www.ccdc.cam.ac.uk/data_request/cif (31 January 2021).

### 2.5. General Methods for Optical Tests

In this process, 5.3 mg (15 μM) of probe **L** was dissolved in 10.00 mL of EtOH solution to prepare a 1.50 mM stock solution. Then, the nitrates of the metal ions, the sodium salt of anions and small amino molecules (Ag^+^, Al^3+^, Cd^2+^, Co^2+^, Cr^3+^, Cu^2+^, Fe^3+^, Hg^2+^, K^+^, Li^+^, Mg^2+^, Na^+^, Ni^2+^, Pb^2+^, Zn^2+^, AcO^−^, Br^−^, C_2_O_4_^2−^, ClO_4_^−^, Cl^−^, CN^−^, CO_3_^2−^, F^−^, H_2_PO_4_^−^, HCO_3_^−^, HSO_3_^−^, HPO_4_^2−^, I^−^, NO^2−^, PO_4_^3−^, S_2_O_3_^2−^, SO_3_^2−^, SO_4_^2−^, GSH, Hcy, H_2_NCONH_2_, Cys) were accurately weighed and dissolved in 10.00 mL of PBS buffer to form 10 mM ion stock solutions. The preparation method of the PBS buffer solution (10 mM) was as follows: 23 g of PBS phosphate buffer powder was weighed and dissolved in 2 L of ultrapure water, and the pH ranged from 7.20 to 7.40.

## 3. Results and Discussion

### 3.1. Synthesis

A new fluorescent probe **L** was obtained from 1-hydroxy-2,4-diformylnaphthalene (compound **1a**, synthesized from 1-naphthol and hexamethylenetetramine) and 1,3,3-trimethyl-2-methyleneindoline by means of a Knoevenagel reaction, as shown in Scheme 1. The molecular structure was characterized by ^1^H NMR spectroscopy, HRMS and single crystal X-ray diffraction. The probe **L** exhibited excellent solubility in common organic solvents (such as methanol, ethanol, DMSO, etc.) and possessed good acid- and alkali-resistance over the pH range 3–11 over 24 h (Appendix A). The thickness of the dish is 1 cm; that is, the thickness of the liquid layer. The concentration of SO_3_^2−^/HSO_3_^−^ was 0.45 μM, and εmax = 17,608.89 L·mol^−1^·cm^−1^, λ(abs) = 550 nm. This work provides a new strategy for the practical application of small molecule probes in the field of anion detection.

### 3.2. Determination of Optimum Experimental Conditions

Anion fluorescent probes are mainly used in the fields of biology, medicine and food monitoring, and so they will have more extensive value if the recognition can be conducted in aqueous solution. In addition, a buffer solution can be used to control the stable pH value in an aqueous solution, making the results of identification more reliable [23,24]. Therefore, the influence of water content on probe **L** was explored by changing the water content during the experiment.

As shown in Figure 1, the fluorescent probe **L** emitted pink emission with λmax em = 605 nm in pure EtOH solution. As the water fraction (*f*w) gradually increased from 0% to 60%, the maximum absorbance and the fluorescence intensity of the probe **L** increased with the increase in the water fraction (*f*w). When the water fraction (*f*w) reached 60%, the absorbance and the fluorescence intensity of the solution attained the maximum value, and the mixture exhibited bright pink light under 365 nm UV irradiation. Then, as the water fraction (*f*w) continued to increase, the fluorescence intensity gradually decreased, and an aggregate-induced quenching process occurred, and the fluorescence quenching efficiency reached 79.95%. Given this, we chose the mixture of EtOH/water (V_EtOH_/V_H2O_ = 2:3) as the recognition environment.

The pH value of the environment is a critical parameter that may affect the selectivity, sensitivity and detection limit of the probe [25]. As shown in Figure 2 and Figure 3, the UV-Vis absorption and fluorescence spectra of probe **L** and the UV-Vis absorption and fluorescence spectra of sulfites/bisulfites (SO_3_^2−^/HSO_3_^−^) identified by probe **L** were experimentally studied over the pH range of 1 to 14.

We added 1.80 mL of PBS buffer solution with different pH values into a 3.00 mL colorimetric dish, and then added 0.03 mL of probe reserve solution. The solution was brought up to a constant volume of 3.00 mL with anhydrous ethanol, shaken well and left to react completely. The influence of different pH values on the probe was measured by UV-Vis spectrophotometer and fluorescence photometer. As shown in Figure 2, in the detection system comprised of EtOH/water (V_EtOH_/V_H2O_ = 2/3, 10 mM PBS buffer), the maximum absorbance of probe **L** is at 550 nm, and the maximum emission peak is at 605 nm over the pH range of 3 to 11. In this wide range, the absorbance and fluorescence intensity of probe **L** are only slightly affected by the pH.

We added 1.80 mL of PBS buffer solution of different pH into a 3.00 mL colorimetric dish, then added 22.50 μL of SO_3_^2−^/HSO_3_^−^ reserve solution and 0.03 mL of probe reserve solution, and used anhydrous ethanol to bring the volume up to 3.00 mL, shook the solution well and left it to stand until the solution was completely reacted. The influence of different pH on the interaction between probe **L** and SO_3_^2−^/HSO_3_^−^ was determined by UV-Vis spectrophotometer and fluorescence photometer. As shown in Figure 3, in the detection system comprising EtOH/water (V_EtOH_/V_H2O_ = 2/3, 10 mM PBS buffer), the maximum absorption peak at 550 nm and the maximum emission peak at 605 nm were reduced by adding SO_3_^2−^/HSO_3_^−^ (750 μM) to the solution of probe **L**, and the spectrum was almost unaffected over the pH range of 3 to 11.

Following the response experiments of water fraction and pH value to probe **L** and the identification and detection of SO_3_^2−^/HSO_3_^−^ with the probe **L**, we chose EtOH/water (V_EtOH_/V_H2O_ = 2/3, 10 mM PBS buffer, pH = 7.40) as the detection system conditions. We also tested the time-dependent optical stability of probe **L** and the **L**-SO_3_^2−^/HSO_3_^−^ mixture, and the results revealed that **L** and the **L**-SO_3_^2−^/HSO_3_^−^ complex responded quickly and were stable over a certain period of time (Appendix A).

### 3.3. Anion Sensing Study

The high selectivity and sensitivity of the probe are key parameters for the detection of domestic water and in vivo studies. Therefore, to test the ability to detect anions, probe **L** (15 μM) was exposed to many anions (such as AcO^−^, Br^−^, C_2_O_4_^2−^, ClO_4_^−^, Cl^−^, CN^−^, CO_3_^2−^, F^−^, H_2_PO_4_^−^, HCO_3_^−^, HSO_3_^−^, HPO_4_^2−^, I^−^, NO^2−^, PO_4_^3−^, S_2_O_3_^2−^, SO_3_^2−^, SO_4_^2−^, [A]^n−^ = 750 μM), metal ions (such as Ag^+^, Al^3+^, Cd^2+^, Co^2+^, Cr^3+^, Cu^2+^, Fe^3+^, Hg^2+^, K^+^, Li^+^, Mg^2+^, Na^+^, Ni^2+^, Pb^2+^, Zn^2+^, [M]^n+^ = 750 μM) and small amino-containing molecules (such as GSH, Hcy, H_2_NCONH_2_, Cys, [M]^n+^ = 750 μM) in mixtures of EtOH and water (V_EtOH_/V_H2O_ = 2/3, pH = 7.40).

As shown in Figure 4, on adding the anions and small amino-containing molecules to the solvent containing **L**, only SO_3_^2−^/HSO_3_^−^ caused the solution’s color to change via naked-eye observation (Appendix A). The absorption spectra and fluorescence spectra of the **L–**anion mixture indicated that probe **L** exhibits good selectivity toward SO_3_^2−^/HSO_3_^−^, while other cations or anions (Appendix A) had little impact on the optical behavior of probe **L**. On other hand, under a 365 nm UV lamp, only the **L**-SO_3_^2−^/HSO_3_^−^ mixture led to the emission light quenching dramatically (Appendix A). Furthermore, competitive experiments were also performed to investigate the selectivity of the probe toward SO_3_^2−^/HSO_3_^−^. When SO_3_^2−^/HSO_3_^−^ was present in the solution, the absorbance of the mixture decreased at 550 nm, and the emission of the mixture at λ_em_ = 605 nm was quenched, while without SO_3_^2−^/HSO_3_^−^, the absorbance and emission barely changed (Figure 5 and Figure 6), which suggested that the coexisting cations/anions/small amino-containing molecules had only a limited impact on the detection of SO_3_^2−^/HSO_3_^−^. Thus, the interference experiments indicated that the probe displays high specificity and selectivity for detecting SO_3_^2−^/HSO_3_^−^ ions.

### 3.4. Titration and Detection Limits

Based on the above experimental conditions, the UV titration experiments were performed with progressive addition of SO_3_^2−^/HSO_3_^−^, and the results are presented in Figure 7. As the figure demonstrates, the absorbance of probe **L** at 550 nm gradually decreased as the SO_3_^2−^/HSO_3_^−^ ions were added. In addition, when the concentration of probe **L** changes from 30 to 300 μM, there exists a good linear relationship between the probe and the SO_3_^2−^/HSO_3_^−^ (y = 0.88828 − 0.02592x, R^2^ = 0.99004). Herein, the detection limit was calculated by utilizing the data of the UV titration experiments following the IUPAC method: 10 groups of blank samples were tested in the absence of sulfite/bisulfite under the same conditions, and then the standard deviation (SD) was calculated from the absorption peak at 550 nm. After that, following the formula: the detection limit = 3SD/S, where S is the slope of the linear relationship during the UV titration, the detection limit of probe **L** for SO_3_^2−^/HSO_3_^−^ is calculated to be 0.24 μM. Compared with other SO_3_^2−^/HSO_3_^−^ probes (Appendix A), the probe **L** has the advantages of a lower detection limit and quicker response time.

As shown in Figure 8, based on the above experimental conditions, the fluorescence titration experiments were performed with progressive addition of SO_3_^2−^/HSO_3_^−^. As the figure demonstrates, the fluorescence intensity of probe **L** at λ_max em_ = 605 nm gradually decreased as the SO_3_^2−^/HSO_3_^−^ ions were added. In addition, when the concentration of probe **L** changed from 15 to 300 μM, there exists a good linear relationship between the probe and the SO_3_^2−^/HSO_3_^−^ ions (y = 350.73493 − 7.35342x, R^2^ = 0.99601). Herein, the detection limit was calculated by utilizing the data of the fluorescence titration experiments following the IUPAC method: 10 groups of blank samples were tested in the absence of sulfite/bisulfite under the same conditions, and then the standard deviation (SD) was calculated from the emission peak at 605 nm. After that, following the formula: the detection limit = 3SD/S, where S is the slope of the linear relationship during the fluorescence titration, the detection limit of probe **L** for SO_3_^2−^/HSO_3_^−^ is calculated to be 9.93 nM. Compared with other SO_3_^2−^/HSO_3_^−^ probes (Appendix A), the probe **L** has the advantages of a lower detection limit and a simpler synthetic route.

### 3.5. A Possible Mechanism for Detection SO_3_^2−^/HSO_3_^−^

The crystal structure shows that the aldehyde group at the 2 position of the 1-hydroxy-2,4-diformylnaphthalene reacts with 1,3,3-trimethyl-2-imethylindoline, but the aldehyde group at the 4 position does not react. The result is the fluorescent probe **L** in which an electron donor (tertiary amine) and acceptor (carbonyl) are connected by a double bond. The indole ring and the naphthalene ring are not in the same plane, and the dihedral angle between them is 164.95°. The bond length of C24-O2 is only 0.1234 nm, indicating that the phenolic hydroxyl group on the naphthalene ring has changed into the ketone structure (as shown in Figure 9).

According to literature reports on the recognition mechanism of SO_3_^2−^/HSO_3_^−^ with fluorescent probes [26,27,28], combined with the above experimental results, it is speculated that the reaction process of probe **L** to recognize SO_3_^2−^/HSO_3_^−^ is as shown in Figure 10.

Due to the influence of two strongly electron-withdrawing carbonyl groups in the probe structure, the electron cloud density of the C=C that connects 1-hydroxy-2,4-diformylnaphthalene and 1,3,3-trimethyl-2-methyleneindoline is not uniform, so it is vulnerable to attack by SO_3_^2−^/HSO_3_^−^ and the addition reaction of C=C occurs, which destroys the original large conjugated structure. With the gradual addition of SO_3_^2−^/HSO_3_^−^, the maximum absorption peak of the UV-Vis absorption spectrum and the strongest fluorescence emission peak of the probe gradually decreased, and the color of the solution gradually became lighter. As shown in Figure 11, the reaction solution of probe **L** and NaHSO_3_ was verified by high resolution mass spectrometry. [C_24_H_24_NO_5_S]^−^: the theoretical value was 437.1302, and the measured value was 437.1262.

### 3.6. Applications

In order to further evaluate the potential application of probe **L** for the detection of SO_3_^2−^/HSO_3_^−^ in real specimens, water samples from an artificial lake (at Guizhou Medical University) and running water (at our laboratory) have been collected for testing. The specific experimental process is as follows: 3.90 mL EtOH solution, 100 μL probe stock solution (15 μM), 3 mL PBS buffer solution and 3 mL water sample (filtered) were added into one volumetric flask and the mixture was shaken well. At the same time, another water sample was processed with the same steps and an appropriate amount of the standard substance (NaHSO_3_) was added. After standing for 2 min., the absorbance at 550 nm and fluorescence intensity at 605 nm of the sample was recorded for further calculations. As shown in Table 1, by UV-Vis absorption spectroscopy, the recoveries of the probe were calculated in the range of 99.20%–104.30% in lake water and 100.01%~104.80% in tap water. As shown in Table 2, by fluorescence spectroscopy, the recoveries of the probe were calculated in the range of 100.50%–108.61% in lake water and 102.72%~103.80% in tap water. These results suggest that **L** is a sensitive and selective probe for SO_3_^2−^/HSO_3_^−^ monitoring in environmental water samples.

## 4. Conclusions

In summary, we have developed a new fluorescent probe based on 1-hydroxy-2,4-diformylnaphthalene. Furthermore, in the presence of SO_3_^2−^/HSO_3_^−^ ions, the probe solution showed an obvious color change from pink to colorless under daylight and from bright to dark under UV lamp irradiation with a detection limit as low as 0.24 μM using UV-Vis absorption spectroscopy and 9.93 nM using fluorescence spectroscopy, respectively. This indicates that the probe **L** has the potential to be used for the detection of SO_3_^2−^/HSO_3_^−^ by the naked eye and via instrumentation. Based on the titration experiments, a good linear relationship was found which allows the probe to be applied to the quantitative and qualitative detection of SO_3_^2−^/HSO_3_^−^ in real samples. We believe that this work not only provides a new example of a small molecular probe for ion detection, but these results may inform researchers in broader fields such as cell imaging, and such research is ongoing in our laboratory.

## Data Availability

No new data were created or analyzed in this study. Data sharing is not applicable to this article.

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
