# Peer review of "A 1-Hydroxy-2,4-Diformylnaphthalene-Based Fluorescent Probe and Its Detection of Sulfites/Bisulfite"

_molecules, 2021, doi:10.3390/molecules26113064_

Round 1

Reviewer 1 Report

The authors prepared a naphthalene-indole conjugated dye for optical detection of sulfite and bisulfite anion. It works well in a wide range of pH in partially aqueous media with good selectively over other cations and anions. However, several important issues need to be addressed:

  1. As shown in Table S4, there are numerous fluorescent probes designed for sulfite detection. Thus, authors must introduce previous work on this and describe notable examples in the introduction part around L54-59.
  2. Does the aldehyde group in probe L react with sulfite? From Table S4, many other similar probes bear aldehyde groups that seem to be the key in the sensing process. Nucleophilic addition of sulfite to the aldehyde also give the same M.W. in mass spectra in Fig.11. Thus, NMR data is needed to confirm the proposed mechanism.
  3. L270-272 and L96, all HRMS data must have 4 decimal places, not 2.
  4. It needs to add a large excess of sulfite (50eq over dye) to reach considerable fluorescent quenching, please comment on this.
  5. In Fig1, there seems to be an aggregate induce quenching process when water was added to the dye in alcohol. Please add explanation.
  6. L302 ‘these results may inform…biometrics and photodynamic therapy’, I don’t see a connection between this work and biometrics and therapy.
  7. Section 2.3, please add protocols to obtain the x-ray crystals.
  8. Section 2.4, add procedures to adjust pH in the sensing process used in Fig2 and 3.
  9. Authors need to make sure the fonts are consistent throughout. Inconsistencies in format are widely seen in Abstract, 2.2 and 2.3.

Author Response

Dear reviewer:

        We have completed our revision, we hope we could address your comment and suggestion.

The authors prepared a naphthalene-indole conjugated dye for optical detection of sulfite and bisulfite anion. It works well in a wide range of pH in partially aqueous media with good selectively over other cations and anions. However, several important issues need to be addressed:

Major Points:

  1. As shown in Table S4, there are numerous fluorescent probes designed for sulfite detection. Thus, authors must introduce previous work on this and describe notable examples in the introduction part around L54-59.

Response: thank you very much for your suggestion, we have introduced the Chinese Standard for the Use of Food Additives on the detection of sulfites.(in the introduction part around L54-59, marked in yellow).

  1. Does the aldehyde group in probe L react with sulfite? From Table S4, many other similar probes bear aldehyde groups that seem to be the key in the sensing process. Nucleophilic addition of sulfite to the aldehyde also give the same M.W. in mass spectra in Fig.11. Thus, NMR data is needed to confirm the proposed mechanism.

Response: Thank you very much for your suggestions. According to you kind suggestions, we had already performed the 1H NMR titration experiment; however, the NMR spectrum was too complicated due to the simultaneous enol form and keto form. As matter of fact, according to the UV titration results, the absorbance at 550 nm gradually decreased after the addition of SO32- anions, which indicated the destruction of the conjugate structure caused by the addition reaction at the olefin groups. Additionally, a large number of references had reported the Nucleophilic addition of sulfite to the olefin groups (Ref. 1-4). Thus, a plausible recognition mechanism was shown in Figure 10.

Reference:

[1] Liu, Y.; Li, K.; Wu, M.; Liu, Y.; Xie, Y.; Yu, X., A Mitochondria-targeted Colorimetric and Ratiometric Fluorescent Probe for Biological SO2 Derivatives in Living Cells. Chemical Communications 2015, 51, 10236-10239.

[2] Xu, J.; Yuan, H.; Zeng, L.; Bao, G., Recent progress in Michael addition-based fluorescent probes for sulfur dioxide and its derivatives. Chinese Chemical Letters 2018, 29 (10), 1456-1464.

[3] Ma, Y.; Tang, Y.; Zhao, Y.; Gao, S.; Lin W., Two-Photon and Deep-Red Emission Ratiometric Fluorescent Probe with a Large Emission Shift and Signal Ratios for Sulfur Dioxide: Ultrafast Response and Applications in Living Cells, Brain Tissues, and Zebrafishes. Analytical Chemistry 2017, 89(17), 9388-9393.

[4] Liu, Y.; Li, K.; Xie, K.; Li, L.; Yu, K.; Wang, X.; Yu, X., A water-soluble and fast-response mitochondria-targeted fluorescent probe for colorimetric and ratiometric sensing of endogenously generated SO2 derivatives in living cells. Chemical Communications 2016, ,52, 3430-3433.

  1. L270-272 and L96, all HRMS data must have 4 decimal places, not 2.

Response: thank you very much for your suggestion, we have corrected the HRMS data to have 4 decimal places.(in page 3 and 11, marked in yellow).

  1. It needs to add a large excess of sulfite (50eq over dye) to reach considerable fluorescent quenching, please comment on this.

Response: Thank you very much for your comments. Generally, it needs to addition sufficient analytes until there's no change of the fluorescence spectra, so that the valid linear range could be observed. During our fluorescent titration experiments, the stable fluorescence intensity was observed until 200 equivalents SO32- anions were added.

  1. In Fig1, there seems to be an aggregate induce quenching process when water was added to the dye in alcohol. Please add explanation.

Response: thank you very much for your suggestion, we have explained the quenching process you mentioned.(in section 3.2, marked in yellow).

  1. L302 ‘these results may inform…biometrics and photodynamic therapy’, I don’t see a connection between this work and biometrics and therapy.

Response: thank you very much for your suggestion, we have corrected the misrepresentation.(in the conclusions part, marked in yellow).

  1. Section 2.3, please add protocols to obtain the x-ray crystals.

Response: thank you very much for your suggestion, we have added protocols to obtain the x-ray crystals.(in section 2.4, marked in yellow).

  1. Section 2.4, add procedures to adjust pH in the sensing process used in Fig2 and 3.

Response: thank you very much for your suggestion, we have added procedures to adjust pH in the sensing process used in Fig2 and 3 in page 5 and 6.( marked in yellow).

  1. Authors need to make sure the fonts are consistent throughout. Inconsistencies in format are widely seen in Abstract, 2.2 and 2.3.

Response: thank you very much for your suggestion, we have corrected the inconsistencies in the font format.(in Abstract, 2.2 and 2.3, marked in yellow).

Reviewer 2 Report

The work is well done and thorough, and it is of significance.  My group is working on a system that is similar the way it detects sulfite and bisulfite, although a completely different molecule and we have done identical experiments and consequent analysis, so I really cannot criticize the (same) approach!  Just to help: could the authors make it clear if they mean Limit of Detection (LOD) as a detection limit, and do they have a Practical Limit of Quantification (PLQ) for their study?

It is nice work and deserves publication.

Author Response

Dear reviewer:

     We have completed our revision, we hope we could address your comment and suggestion.

The work is well done and thorough, and it is of significance.  My group is working on a system that is similar the way it detects sulfite and bisulfite, although a completely different molecule and we have done identical experiments and consequent analysis, so I really cannot criticize the (same) approach!  Just to help: could the authors make it clear if they mean Limit of Detection (LOD) as a detection limit, and do they have a Practical Limit of Quantification (PLQ) for their study?

  1. Just to help: could the authors make it clear if they mean Limit of Detection (LOD) as a detection limit, and do they have a Practical Limit of Quantification (PLQ) for their study?

Response: thank you very much for your question, the study had a Practical Limit of Quantification (PLQ) of 2 mg/kg, which was converted to a LOD of 4.18x10-5 mol/L.

Yours,

Sincerely.

Reviewer 3 Report

The manuscript by Shi et al. presents the synthesis of a novel compound to probe the presence of sulfite/bisulfite anions in real water samples. The detection properties of the probe are interesting, and could be of interest to the community. However, in the current state the manuscript lacks a few experimental evidences regarding characterizations and statement about the sulfonated species.

Given the nature and the scope of the journal, full characterization of the compounds is critical. Compound 1a and the final compound L are not fully characterized, and the 1H NMR spectra are not provided in the SI. I would like the authors to include the 13C NMR shifts in the experimental protocol and to provide the spectra in the supporting information (for 1a and L). Additionally, in the characterization section, the authors can include the UV-vis measurements: λ(abs) and ε for compound L. 

In Figure 10, the authors state that the sulfonated species is as indicated. However, experimental evidence is missing... Please provide NMR evidence to support the claim.

Minor correction:

Line 88: Synthesis of compound L should be section 2.3 and all the other sections need to be revised accordingly.

Line 81-87: Please remove the italic font.

Line 264: electron-adsorbing, revise as electron-withdrawing

Line 265: .... of the C=C bond, please be more specific about which C=C bond you are refering to. 

Author Response

Dear reviewer:

     We have complete our revision, we hope we could address your comment and suggestion. 

The manuscript by Shi et al. presents the synthesis of a novel compound to probe the presence of sulfite/bisulfite anions in real water samples. The detection properties of the probe are interesting, and could be of interest to the community. However, in the current state the manuscript lacks a few experimental evidences regarding characterizations and statement about the sulfonated species.

  1. Given the nature and the scope of the journal, full characterization of the compounds is critical. Compound 1a and the final compound L are not fully characterized, and the 1H NMR spectra are not provided in the SI. I would like the authors to include the 13C NMR shifts in the experimental protocol and to provide the spectra in the supporting information (for 1a and L). Additionally, in the characterization section, the authors can include the UV-vis measurements: λ(abs) and ε for compound L. 

Response: thank you very much for your suggestion, we have added the 1H NMR and 13C NMR spectra in the SI, UV-vis measurements in section 3.1.(marked in green).

  1. In Figure 10, the authors state that the sulfonated species is as indicated. However, experimental evidence is missing... Please provide NMR evidence to support the claim.

Response:Thank you very much for your comments. We had provided the mass spectrometry (MALDI-TOF) to investigate the complexation effect. When the SO32- anions were added to the solution of L, a new peak appeared at m/z 437.1262, corresponding to a new formed SO32—L complex (please see Figure 11 in the text). This result strongly confirms the SO32- anions addition to the Ligand.

Minor correction:

  1. Line 88: Synthesis of compound L should be section 2.3 and all the other sections need to be revised accordingly.

Response: thank you very much for your suggestion, we have corrected the mismarked sections.(in sections 2.3, 2.4 and 2.5, marked in green).

  1. Line 81-87: Please remove the italic font.

Response: thank you very much for your suggestion, we have removed the incorrectly marked italics.(in section 2.3, marked in green).

  1. Line 264: electron-adsorbing, revise as electron-withdrawing

Response: thank you very much for your suggestion, we have corrected the wrong words.(in section 3.5, marked in green).

  1. Line 265: .... of the C=C bond, please be more specific about which C=C bond you are refering to. 

Response: thank you very much for your suggestion, we have indicated the position of the double bond in question.(in section 3.5, marked in purple).

Yours,

Sincerely.

Round 2

Reviewer 1 Report

Given that the authors have addressed most of my concerns in the initial review, the manuscript can now be accepted in the present form. 

Reviewer 3 Report

The authors have addressed the major and minor points highlighted previously. Though the work does not represent an important level of novelty, I think it is relevant to the journal's scope and can be of interest to the wider community.